# EFFICIENT UNIFIED MULTIMODAL UNDERSTANDING AND GENERATION WITH GATED HYBRID ATTENTION

## ABSTRACT

Unified multimodal learning requires attention mechanisms that are both efficient and expressive. Softmax attention provides strong modeling capacity but suffers from quadratic complexity, while linear attention achieves near-linear efficiency at the cost of weaker expressivity. We identify two major expressivity challenges in efficient unified multimodal models: (*i*) modality imbalance, where dominant signals suppress weaker modalities during fusion, and (*ii*) loss of global context, as efficient variants tend to over-smooth long sequences. We propose **Gated Hybrid Attention (GHA)**, a multimodal-specialized operator that augments linear attention with (*i*) a selective gating mechanism to balance modality contributions and stabilize training, and (*ii*) agent-token softmax aggregation to restore adaptive global context while preserving near-linear complexity. To demonstrate generality, we validate GHA in two representative paradigms: autoregressive-only(AR-only) and autoregressive+diffusion(AR+Diffusion). In both settings, GHA consistently improves multimodal alignment, long-context retention, and efficiency over comparable Transformer and efficient attention baselines. These cross-paradigm results highlight that GHA functions as a plug-and-play building block, offering a lightweight and extensible approach that is orthogonal to scaling trends and modality complexity.

## 1 INTRODUCTION

Recent advances in Large Language Models (LLMs) Vaswani et al. (2017); Brown et al. (2020); Chowdhery et al. (2022); OpenAI et al. (2024) have spurred interest in extending their capabilities beyond text, leading to the emergence of Multimodal Large Language Models (MLLMs) Caffagni et al. (2024); Wang et al. (2024a); Zhang et al. (2024). Early efforts such as Flamingo Alayrac et al. (2022), GIT Wang et al. (2022a), and OFA Wang et al. (2022b) demonstrated the feasibility of joint vision–language modeling but were often specialized for narrow tasks, motivating the development of more unified architectures. Recent unified systems including Show-o Xie et al. (2024a), JanusFlow Ma et al. (2025), Janus-Pro Chen et al. (2025) and VILA-U Wu et al. (2025) extend Transformer-based pipelines to support both multimodal understanding and generation. However, these models inherit the quadratic complexity of self-attention, which poses scalability bottlenecks for high-resolution inputs and long multimodal sequences.

An alternative line of work replaces Transformers with state-space architectures such as Mamba, exemplified by OmniMamba Zou et al. (2025). While these approaches achieve linear efficiency in both computation and memory, their strictly sequential modeling and limited cross-layer interactions constrain the ability to capture complex long-range dependencies, resulting in weaker expressivity compared to Transformer counterparts.

In contrast, our work addresses the complementary challenge of balancing efficiency and expressivity through architectural design. We introduce *Gated Hybrid Attention (GHA)*, a lightweight and theoretically grounded drop-in substitute for standard Transformer attention, tailored for unified multimodal models. Unlike approaches that rely on large-scale data scaling or specialized backbones, GHA directly enhances the attention mechanism itself, extending linear attention with two key components: (i) a gating mechanism that stabilizes optimization and alleviates modality

imbalance, and (ii) a softmax-based agent-token bottleneck that reintroduces global context while maintaining near-linear complexity. This architectural advance complements large-scale scaling and complex unified backbones, offering a practical direction toward building multimodal systems that are both efficient and expressive.

Our contributions can be summarized as follows:

- **Gated Hybrid Attention.** We introduce GHA, which extends linear attention with a key–value gating module and a softmax-based agent-token bottleneck. This design significantly enhances expressivity, long-context reasoning, and cross-modal integration, while preserving near-linear computational complexity.
- **Hardware-efficient Triton Kernel.** To ensure the theoretical efficiency of GHA translates into practice, we provide a hardware-optimized implementation based on FlashAttention-style chunking and a custom Triton kernel.
- **Empirical Evaluation.** We evaluate GHA on both AR-only and AR+Diffusion unified multimodal pipelines. Results show consistent gains in multimodal alignment, long-context retention, and generation quality compared to Transformer and efficient attention baselines. These cross-paradigm results highlight that GHA functions as a plug-and-play building block, complementary to large-scale scaling and specialized backbones.

## 2    RELATED WORK

### 2.1    UNIFIED MULTIMODAL MODELS

Early frameworks such as Flamingo, GIT, and OFA demonstrated the feasibility of coupling vision and language but were limited to narrow task formats. Subsequent research has sought fully unified architectures that support both multimodal understanding and generation within a single backbone. As summarized in Show-o Xie et al. (2024a), existing systems fall into two paradigms: *AR-only*, where all tasks are cast as next-token prediction (e.g., Emu3 Wang et al. (2024c), VILA-U Wu et al. (2025)); and *AR+Diffusion*, where an AR backbone is augmented with a discrete diffusion module for higher-fidelity generation (e.g., Show-o Xie et al. (2024a), JanusFlow Ma et al. (2025), Janus-Pro Chen et al. (2025)). Concurrent works such as Show-o2 Xie et al. (2025) and BAGEL Deng et al. (2025) push performance further by scaling data and adopting increasingly sophisticated multimodal backbones (e.g., decoupled visual encoders, 3D causal latent spaces, mixture-of-transformer-experts). While these systems substantially improve multimodal generation quality, they also reveal persistent issues of cross-modal imbalance, where dominant modalities (e.g., text) suppress weaker ones during fusion. Moreover, most progress has come from scaling data, model size, and backbone complexity, leaving the core attention operator largely unchanged.

### 2.2    EFFICIENT SEQUENCE MODELING

To mitigate the quadratic cost of attention, linear variants (e.g., Linear Transformer Katharopoulos et al. (2020), Performer Choromanski et al. (2020), Reformer Kitaev et al. (2020)) approximate softmax kernels or restrict interactions, reducing complexity to linear time and memory. In parallel, state-space models such as Mamba Gu & Dao (2024) and Mamba-2 Dao & Gu (2024) achieve similar efficiency through recurrent parameterizations, while hybrid designs like Jamba Datta et al. (2024) combine state-space recurrence with sparse attention. While these architectures excel in efficiency, they often struggle to maintain global context in long sequences and thus sacrifice expressivity. In particular, approximations in linear attention can lead to over-smoothing, while strictly recurrent models limit cross-token interactions across layers. As a result, efficient sequence modeling provides strong scalability but remains insufficient for tasks that require rich long-range dependencies.

### 2.3    EXPRESSIVITY ENHANCEMENTS

Several works aim to strengthen attention capacity. Gated designs such as GLA Yang et al. (2024) and GTrXL Parisotto et al. (2019) improve stability and selectivity by modulating key–value or residual pathways. Another family of approaches strengthens expressivity through *agent tokens*,

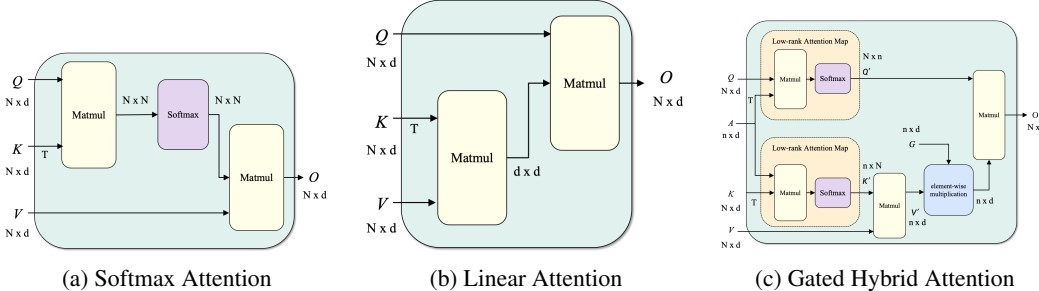

(a) Softmax Attention      (b) Linear Attention      (c) Gated Hybrid Attention

Figure 1: Difference between softmax attention ($\mathcal{O}(N^2 d)$), linear attention ($\mathcal{O}(Nd^2)$), and GHA ($\mathcal{O}(N\,n\,d)$).

where a small set of latent units mediate communication between input tokens. Representative examples include Set Transformer Lee et al. (2019) with inducing points, Perceiver and Perceiver IO Jaegle et al. (2021; 2022) with latent bottleneck arrays, and Agent Attention Wang et al. (2024b) with explicitly defined agent tokens. These agents compress token interactions into a compact latent space and then redistribute information back to the sequence, typically using content-dependent weighting such as softmax aggregation. While effective, such designs have been studied mainly in unimodal contexts and remain underexplored in unified multimodal systems. Overall, unified multimodal models have advanced primarily through scaling data and backbone complexity, with some architectural innovations targeting specialized multimodal components. In parallel, efficient sequence modeling has been explored mostly in unimodal contexts and remains rarely applied to unified multimodal systems. As a result, the attention operator at the core of unified models is still underexplored, facing persistent challenges of cross-modal imbalance (dominant modalities suppressing weaker ones), loss of global context in long sequences, and quadratic computational complexity that limits scalability. To address these challenges, we propose GHA, which extends linear attention with gating and agent-token–based aggregation. This balances efficiency and expressivity, yielding improvements across both AR-only and AR+Diffusion paradigms. This architectural direction is complementary to scaling- and backbone-oriented advances and points toward future integration into more complex unified frameworks.

## 3 PRELIMINARY

**Softmax Attention.** Given queries $\mathbf{Q}$, keys $\mathbf{K}$, and values $\mathbf{V}$, standard attention computes

$$\mathbf{O} = \mathrm{Softmax}\left(\frac{\mathbf{Q}\mathbf{K}^\top}{\sqrt{d}}\right)\mathbf{V}, \tag{1}$$

which provides strong expressivity but incurs $\mathcal{O}(N^2 d)$ complexity for sequence length $N$ and hidden dimension $d$.

**Linear Attention.** To alleviate this bottleneck, Linear Attention Katharopoulos et al. (2020) reformulates the exponential similarity $\exp(q^\top k)$ as a kernelized dot-product $\phi(q)^\top \phi(k)$, where $\phi(\cdot)$ is a positive feature mapping (e.g., $\phi(x) = \mathrm{ELU}(x) + 1$ or randomized Fourier features). This enables computing attention as

$$\mathbf{O} \approx \phi(\mathbf{Q})\big(\phi(\mathbf{K})^\top \mathbf{V}\big), \tag{2}$$

which reduces complexity to $\mathcal{O}(Nd^2)$ and admits an equivalent recurrent form for streaming updates. In practice, however, many implementations simplify $\phi(\cdot)$ to the identity mapping, which abandons strict kernel approximation but often yields more stable and competitive performance in downstream tasks Sun et al. (2023).

Our proposed GHA builds on Linear Attention by (i) incorporating a gating mechanism to dynamically filter memory updates, and (ii) reintroducing content-dependent softmax weighting through agent tokens, which aggregate and redistribute global context. These additions retain linear-time efficiency while substantially improving expressivity and stability in long-sequence modeling.

## 4  GATED HYBRID ATTENTION

As shown in Figure 1, we propose a three-stage hybrid architecture, termed GHA. This model augments linear attention through two key components: a selective gating mechanism applied to the key-value pathway and a softmax-based aggregation over agent tokens. The proposed design effectively addresses the two primary challenges in unified multimodal modeling: efficiency, attained by maintaining linear computational complexity, and expressivity, enhanced by the introduced agent-token aggregation and gating mechanism.

### 4.1  PARALLEL FORMULATION

**(1) Agent-token based Softmax Aggregation.** To avoid the quadratic query–key interactions, GHA introduces a small set of $n \ll N$ agent tokens $\mathbf{A} \in \mathbb{R}^{n \times d}$, which serve as an information bottleneck that aggregates signals from all tokens before redistributing them. All tokens send their query and key representations to these agents via softmax attention:

$$\mathbf{K}' = \mathrm{Softmax}(\mathbf{A}\mathbf{K}^\top) \in \mathbb{R}^{n \times N}, \qquad \mathbf{Q}' = \mathrm{Softmax}(\mathbf{Q}\mathbf{A}^\top) \in \mathbb{R}^{N \times n}. \tag{3}$$

Compared to standard softmax attention, our design preserves content-dependent weighting while reducing the complexity from $\mathcal{O}(N^2)$ to $\mathcal{O}(Nn)$. Compared to plain linear attention, it reintroduces softmax aggregation in a compressed latent space, thereby restoring adaptive global context modeling.

**(2) Linear Accumulation.** Following the practical form of linear attention, we adopt the identity mapping $\phi(x) = x$, which is widely used for its stability. During training, the attention output can be computed in parallel as

$$\mathbf{O} = \mathbf{Q}' \left( \mathbf{K}' \mathbf{V} \right), \tag{4}$$

here $(\mathbf{K}'\mathbf{V}) \in \mathbb{R}^{n \times d}$ and the computation costs $\mathcal{O}(Nnd)$. Compared to softmax attention, this accumulation is more efficient; in GHA we incorporate gating and agent-token based softmax aggregation to mitigate over-smoothing and stabilize long-sequence modeling.

**(3) Selective Gating.** To further stabilize training and suppress over-smoothing, GHA applies gating directly on the aggregated key–value pathway. Formally, we compute

$$\mathbf{O} = \mathbf{Q}' \left( \mathbf{G} \odot (\mathbf{K}'\mathbf{V}) \right), \tag{5}$$

where $\mathbf{G} \in (0,1)^{n \times d}$ is a learnable, data-dependent gating matrix applied to the compressed key–value representation, and $\odot$ denotes element-wise multiplication. To make this data-dependent nature explicit and avoid interpreting $\mathbf{G}$ as a static parameter matrix, we write it as $\mathbf{G} = f(x)$, where $f(x)$ is an input-conditioned projection that produces the dynamic gating weights. This design keeps the parallel complexity at $\mathcal{O}(Nnd)$, while enabling selective control over memory updates. By filtering noisy or redundant updates, gating improves long-sequence stability and prevents modality dominance, a common challenge in unified multimodal training, thus yielding more fine-grained and robust cross-modal alignment.

Together, these designs yield an overall complexity of $\mathcal{O}(Nnd)$ in theory, substantially lower than standard softmax attention ($\mathcal{O}(N^2d)$) while significantly enhancing the representational power of plain linear attention.

### 4.2  RECURRENT FORMULATION

For autoregressive decoding, GHA admits a recurrent formulation that achieves true linear-time complexity during inference as shown in Figure 2. At each time step $t$, the compressed key–value state $\mathbf{S}_t \in \mathbb{R}^{n \times d}$ is updated as

$$\mathbf{S}_t = \mathbf{g}_t \odot \mathbf{S}_{t-1} + \mathbf{k}'_t \mathbf{v}_t^\top, \qquad \mathbf{o}_t = \mathbf{q}'_t \mathbf{S}_t, \tag{6}$$

where $\mathbf{g}_t \in (0,1)^n$ is a token-wise gating vector (broadcast along the $d$ dimension), $\mathbf{k}'_t, \mathbf{q}'_t \in \mathbb{R}^n$ are the agent-projected key and query (with $q'_t$ and $k'_t$ denoting the $t$-th row of $Q'$ and the $t$-th column of $K'$, respectively), and $\mathbf{v}_t \in \mathbb{R}^d$ is the value.

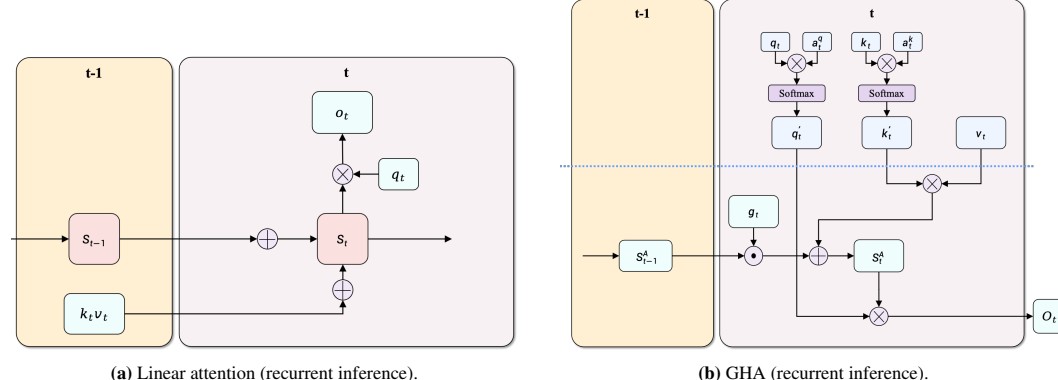

**(a)** Linear attention (recurrent inference).  **(b)** GHA (recurrent inference).

Figure 2: Comparison of (a) linear attention (recurrent inference) and (b) GHA (recurrent inference).

Here $\mathbf{k}_t' \mathbf{v}_t^\top \in \mathbb{R}^{n \times d}$ matches the dimension of $\mathbf{S}_t$. Since each update depends only on $\mathbf{S}_{t-1}$ and the current token $(\mathbf{k}_t', \mathbf{v}_t)$, this recurrent inference form scales linearly with sequence length as $\mathcal{O}(Nnd)$.

During training, causal masking prohibits direct use of the recurrent update. As with other linear attention mechanisms Katharopoulos et al. (2020); Yang et al. (2024); Ma et al. (2021), the masked parallel form requires handling prefix-restricted interactions, which breaks the associativity trick in linear accumulation and leads to quadratic $\mathcal{O}(N^2 d)$ complexity.

### 4.3 HARDWARE-EFFICIENT IMPLEMENTATION

Inspired by FlashAttention Dao et al. (2022) and FlashLinearAttention Yang et al. (2023), we design a GPU-friendly chunkwise execution strategy implemented in Triton to bridge the gap between theoretical efficiency and practical training.

Let the sequence of length $N$ be partitioned into $L = N/C$ contiguous chunks of size $C$. Under causal masking, we treat *inter-chunk* and *intra-chunk* dependencies differently.

**Notation.** We denote by $t \in \{1, \ldots, N\}$ the *token index* along the input sequence of length $N$, and by $i \in \{0, \ldots, L-1\}$ the *chunk index* when the sequence is partitioned into $L = N/C$ contiguous chunks of size $C$. Accordingly, $\mathbf{S}_t$ refers to the recurrent state at token $t$ in the inference (sequential) formulation, while $\mathbf{S}_{\text{start}}^{(i)}$ and $\mathbf{S}_{\text{end}}^{(i)}$ denote the carried states across chunks in the chunkwise (parallel training) formulation.

**Inter-chunk dependencies (no mask).** Across chunks, all tokens in chunk $i+1$ can fully attend to preceding chunks $\{0, \ldots, i\}$ without causal masking. We maintain a carried key–value state $\mathbf{S}^{(i)} \in \mathbb{R}^{n \times d}$ updated as

$$\mathbf{S}^{(i+1)} = \Gamma^{(i+1)} \odot \mathbf{S}^{(i)} + \Delta^{(i+1)}, \qquad \mathbf{O}_{\text{inter}}^{(i+1)} = \mathbf{Q}'[i+1]\, \mathbf{S}^{(i+1)}, \tag{7}$$

where $\Gamma^{(i+1)} \in (0,1)^{n \times d}$ encodes the cumulative decay across chunk $i+1$, and

$$\Delta^{(i+1)} = \sum_{t \in \text{chunk}(i+1)} \mathbf{k}_t' \mathbf{v}_t^\top \in \mathbb{R}^{n \times d}.$$

This component is fully parallelizable across chunks, with total cost $\mathcal{O}(Nnd)$.

**Intra-chunk dependencies (causal mask).** Within each chunk of size $C$, local causality is enforced by applying a lower-triangular mask $\mathbf{M} \in \{0,1\}^{C \times C}$ to restrict interactions to valid prefixes:

$$\mathbf{K}' = \text{Softmax}(\mathbf{A}\mathbf{K}^\top), \qquad \mathbf{Q}' = \text{Softmax}(\mathbf{Q}\mathbf{A}^\top). \tag{8}$$

$$\mathbf{O}_{\text{intra}} = \left(\mathbf{Q}'\mathbf{K}'^\top \odot \mathbf{M}\right)\mathbf{V}. \tag{9}$$

Naively, applying the causal mask incurs $\mathcal{O}(C^2 d)$ cost per chunk, as each token depends on all its predecessors.

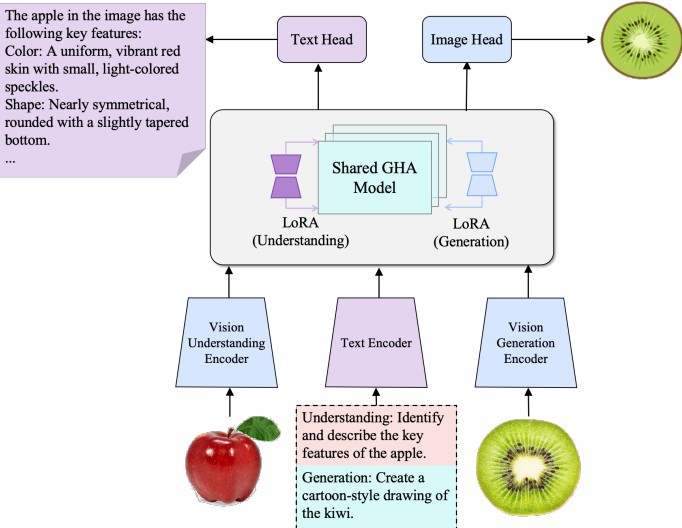

Figure 3: **Architecture of OmniGHA (AR-only).** The system processes three types of inputs: (i) visual features from the *vision understanding encoder*, (ii) visual tokens from the *vision generation encoder*, and (iii) text embeddings from the *text encoder*. All modalities are fed into a shared *Gated Hybrid Attention (GHA)* Transformer decoder, which serves as the core of the architecture. Two modality-specific output heads are employed: a *text head* for understanding tasks and an *image head* for generation tasks, respectively. Task-specific LoRA adapters are applied within the decoder to enable parameter-efficient, modality-specific adaptation while keeping the decoder weights largely shared across tasks.

**Chunkwise combination.** The final outputs combine both contributions:

$$\mathbf{O} = \sum_{i=0}^{L-1} \left( \mathbf{O}_{\text{inter}}^{(i)} + \mathbf{O}_{\text{intra}}^{(i)} \right). \tag{10}$$

Most FLOPs lie in the inter-chunk part (unmasked matmuls over the small agent dimension $n \ll N$), while the intra-chunk part is confined to local $C \times C$ blocks. This decomposition keeps the dominant work parallelizable and yields training that closely approaches the theoretical $\mathcal{O}(Nnd)$ scaling despite causal masking. The persistent state buffers scale as $\mathcal{O}(nd)$ per layer, with $\mathcal{O}(Cnd)$ temporary tiles during chunked training.

### 4.4 OmniGHA: Unified Multimodal Architecture

While GHA is a general attention mechanism, we demonstrate its effectiveness by instantiating it within a simplified unified multimodal pipeline, which we call *OmniGHA*. As shown in Figure 3, we illustrate OmniGHA in its basic AR-only instantiation for clarity. Frozen text and vision encoders provide embeddings, a shared GHA decoder performs multimodal fusion, and lightweight LoRA adapters and task-specific heads produce outputs. This design concentrates most parameters and compute in the shared backbone, highlighting the plug-and-play nature of GHA. A detailed breakdown of modules and the progressive training schedule is provided in Appendix A.1 and Appendix A.2, and we also describe an AR+Diffusion variant in Appendix A.3, where the shared decoder is coupled with a latent predictor and a diffusion-style generation head (e.g., flow matching). This confirms that GHA extends naturally to pipelines beyond AR-only.

To isolate the architectural contribution of GHA, we first evaluate it in the most basic AR-only setting, where frozen encoders feed embeddings into a single shared decoder and lightweight heads handle downstream tasks. This controlled setup deliberately excludes diffusion modules or sophisticated backbones, ensuring that observed gains can be attributed directly to GHA. We then extend OmniGHA to the AR+Diffusion paradigm by replacing the image head with a latent predictor and diffusion head, validating that GHA integrates seamlessly into more complex generation pipelines. The diffusion pathway is kept entirely unchanged to ensure a fair comparison. Across both paradigms, GHA delivers consistent improvements in multimodal alignment, long-context retention, and generation quality over strong Transformer baselines. These results demonstrate that GHA is

| Type | Model | LLM Params | Res. | POPE↑ | MME-P↑ | VQAv2$_{test}$↑ | GQA↑ | MMMU↑ |
|------|-------|-----------|------|-------|--------|-----------------|------|-------|
| Und. Only | LLaVA-Phi Zhu et al. (2024) | Phi-2-2.7B | 336 | 85.0 | 1335.1 | 71.4 | - | - |
| | LLaVA Liu et al. (2024c) | Vicuna-7B | 224 | 76.3 | 809.6 | - | - | - |
| | Emu3-Chat Wang et al. (2024c) | 8B from scratch | 512 | 85.2 | - | 75.1 | 60.3 | 31.6 |
| | LLaVA-v1.5 Liu et al. (2024b) | Vicuna-13B | 448 | 86.3 | 1500.1 | 81.8 | 64.7 | - |
| | InstructBLIP Dai et al. (2023) | Vicuna-13B | 224 | 78.9 | 1212.8 | - | 49.5 | - |
| | MobileVLM Chu et al. (2023) | MobileLLaMA-1.4B | 336 | 84.5 | 1196.2 | - | 56.1 | - |
| | MobileVLM-V2 Chu et al. (2024) | MobileLLaMA-1.4B | 336 | 84.3 | 1302.8 | - | 59.3 | - |
| | LLaVA-v1.5-Phi-1.5 Xie et al. (2024b) | Phi-1.5-1.3B | 336 | 84.1 | 1128.0 | 75.3 | 56.5 | 30.7 |
| Unified | LWM Liu et al. (2024a) | LLaMA2-7B | 256 | 75.2 | - | 55.8 | 44.8 | - |
| | Chameleon Team (2024) | 7B from scratch | 512 | - | - | - | - | 22.4 |
| | LaVIT Jin et al. | 7B from scratch | 256 | - | - | 66.0 | 46.8 | - |
| | Emu3 Wang et al. (2024c) | 8B from scratch | 512 | 85.2 | 1243.8 | 75.1 | 60.3 | 31.6 |
| | Janus Wu et al. (2024) | DeepSeek-LLM-1.3B | 384 | 87.0 | 1338.0 | 77.3 | 59.1 | 30.5 |
| | JanusFlow Ma et al. (2024) | DeepSeek-LLM-1.3B | 384 | 88.0 | 1333.1 | 79.8 | 60.3 | 29.3 |
| | Show-o Xie et al. (2024b) | Phi-1.5-1.3B | 512 | 80.0 | 1097.2 | 69.4 | 58.0 | 26.7 |
| | OmniMamba Zou et al. (2025) | Mamba-2-1.3B | 384 | 86.3 | 1290.6 | 77.7 | 60.8 | 30.6 |
| | **OmniGHA (AR)** | 1.3B from scratch | 384 | **88.7** | **1342.5** | **80.6** | **62.4** | **32.8** |
| | **OmniGHA (AR+Diffusion)** | 1.3B from scratch | 384 | **89.6** | **1354.9** | **81.3** | **62.1** | **33.6** |

Table 1: **Performance comparison on multimodal understanding benchmarks.** "Und. only" models are trained for understanding only; "Unified" models support both understanding and generation.

effective not only in controlled AR-only pipelines but also transferable to AR+Diffusion settings, providing a lightweight building block orthogonal to scaling trends and backbone complexity.

## 5 EXPERIMENTS

### 5.1 DATA AND TRAINING SETUP

We train a 1.3B-parameter model from a scratch attention mechanism. Images are $384 \times 384$ for understanding and $256 \times 256$ for generation via a VQ-VAE tokenizer. We use BF16 AdamW and lightweight rank-8 LoRA on input projectors. Unified training follows a three-phase curriculum: (1) representation warm-up on understanding, (2) long-context adaptation on generation, and (3) joint alignment. We deliberately avoid sophisticated visual stacks to isolate the effect of replacing attention with GHA. Full datasets and hyperparameters are in Appendix A.4.

Recent concurrent works such as Omni-Qwen Xu et al. (2025), BAGEL, and Show-o2 advance unified multimodal learning primarily through larger-scale datasets and increasingly sophisticated backbones (e.g., causal 3D latent spaces, mixture-of-experts, or refined training strategies). Our work instead focuses on an orthogonal dimension: enhancing the attention operator itself. To clearly isolate this effect, we deliberately maintain a controlled evaluation setting rather than comparing against scaling- or backbone-heavy systems.

### 5.2 EVALUATION RESULTS

**Multimodal Understanding.** We follow Show-o, JanusFlow, and OmniMamba, and evaluate on standard multimodal understanding benchmarks, including POPE Li et al. (2023), MME Fu et al. (2024), VQAv2 Goyal et al. (2017), GQA Hudson & Manning (2019), and MMMU Yue et al. (2024). As shown in Table 1, OmniGHA consistently improves over models of similar scale (∼1.3B), achieving 88.7 on POPE, 1342.5 on MME, 80.6 on VQAv2, 62.4 on GQA, and 32.8 on MMMU. It surpasses understanding-only systems such as LLaVA-v1.5-Phi-1.5 (84.1/1128.0/75.3/56.5/30.7) and MobileVLM-V2 (84.3/1302.8/–/59.3/–), while also outperforming unified baselines like Show-o (80.0/1097.2/69.4/58.0/26.7) and JanusFlow (88.0/1333.1/79.8/60.3/29.3), despite JanusFlow being trained on over 65M image–text pairs. Interestingly, the AR+Diffusion variant achieves slightly higher understanding scores, which we attribute to the additional conditioning projector acting as a form of representation regularization during joint training.

**Visual Generation.** We evaluate on MS-COCO Lin et al. (2014) and compare with both generation-only diffusion systems and unified multimodal models (Table 2). Large-scale diffusion models (e.g., DALL·E 2 Ramesh et al. (2022), Imagen Saharia et al. (2022)) progressively reduce

FID to the 7–10 range, with recent refinements such as U-ViT Bao et al. (2023) reaching 5.95. Among unified models, early approaches (CoDI, SEED-X, LWM) lag far behind, while Show-o narrows the gap (9.24). OmniMamba achieves 5.36 when trained on 35M pairs. Our OmniGHA further improves to 5.12, surpassing OmniMamba at the same scale. While the AR+Diffusion variant incurs additional cost, it still surpasses Show-o and JanusFlow in FID, demonstrating that GHA can transfer effectively to diffusion pipelines while retaining high generation quality.

Please note that our paper does not claim SOTA over all understanding- or generation-only models. The contribution lies in the GHA module—an attention operator that provides a more stable and efficient architecture under the unified multimodal setting. For fairness, we use our own Transformer baseline trained from scratch under the same data, compute, and training pipeline as OmniGHA. Additional details are provided in Appendix A.5.

| Model | Inference Time (s) | | | | | |
| | 4K | 8K | 16K | 32K | 64K | 128K |
|---|---|---|---|---|---|---|
| Show-o Xie et al. (2024b) | 153 | 218 | 289 | OOM | OOM | OOM |
| JanusFlow Ma et al. (2024) | 91 | 125 | 153 | 182 | 223 | 268 |
| OmniMamba Zou et al. (2025) | 16 | 43 | 85 | 127 | 153 | 187 |
| **OmniGHA (AR)** | **12** | **38** | **75** | **110** | **132** | **156** |
| **OmniGHA (AR+Diffusion)** | **13** | **36** | **79** | **113** | **128** | **151** |

Table 3: **Inference efficiency on multimodal understanding.** OmniGHA (AR-only) achieves the best scalability across long contexts, consistently outperforming both Transformer- and Mamba-based baselines. The AR+Diffusion variant likewise surpasses all baseline models across comparable sequence lengths, underscoring its versatility across paradigms. OOM: Out-of-Memory

| Model | Speed (Image/s) | Time (s) |
|---|---|---|
| Show-o Xie et al. (2024b) | 0.81 | 19.66 |
| JanusFlow Ma et al. (2024) | 1.02 | 15.64 |
| OmniMamba Zou et al. (2025) | 5.68 | 2.81 |
| **OmniGHA (AR)** | **5.26** | **2.97** |
| **OmniGHA (AR+Diffusion)** | **1.95** | **17.2** |

Table 4: **Visual generation efficiency.** OmniGHA (AR-only) approaches OmniMamba while being $5\times$ faster than Show-o and Janus-Flow. The AR+Diffusion variant is slower, reflecting the inherent overhead of diffusion, but demonstrates that GHA remains compatible and effective when integrated into heavier generative pipelines.

## 5.3 EFFICIENCY RESULTS

We compare the inference efficiency of OmniGHA with prior Transformer- and Mamba-based unified models on both multimodal understanding and visual generation, using a single NVIDIA A100 GPU in FP16.

For understanding (Table 3), OmniGHA (AR-only) achieves the lowest latency across sequence lengths up to 128k. At 32k and 128k tokens, it is $1.15\times$–$1.2\times$ faster than OmniMamba, and significantly outperforms Show-o ($12.8\times$) and JanusFlow ($2.3\times$). The AR+Diffusion variant also outperforms all baseline systems across comparable lengths, demonstrating that GHA preserves scalability and efficiency even when integrated with diffusion-style modules. Overall, these results confirm that both OmniGHA variants are competitive solutions for long-context multimodal understanding.

| Type | Model | Params | Images | FID-30K↓ |
|---|---|---|---|---|
| Gen. Only | DALL·E Ramesh et al. (2021) | 12B | 250M | 27.5 |
| | GLIDE Nichol et al. (2021) | 5B | 250M | 12.24 |
| | DALL·E 2 Ramesh et al. (2022) | 6.5B | 650M | 10.39 |
| | SDv1.5 Rombach et al. (2022) | 0.9B | 2000M | 9.62 |
| | PixArt Chen et al. (2023) | 0.6B | 25M | 7.32 |
| | Imagen Saharia et al. (2022) | 7B | 960M | 7.27 |
| | Parti Yu et al. (2022) | 20B | 4.8B | 7.23 |
| | Re-Imagen Chen et al. (2022) | 2.5B | 50M | 6.88 |
| | U-ViT Bao et al. (2023) | 45M | 83K(coco) | 5.95 |
| Unified | CoDI Tang et al. (2024) | - | 400M | 22.26 |
| | SEED-X Ge et al. (2024) | 17B | - | 14.99 |
| | LWM Liu et al. (2024a) | 7B | - | 12.68 |
| | DreamLLM Dong et al. (2023) | 7B | - | 8.76 |
| | Show-o Xie et al. (2024b) | 1.3B | 35M | 9.24 |
| | OmniMamba Zou et al. (2025) | 1.3B | 83K(coco) | 5.50 |
| | OmniMamba | 1.3B | 35M | 5.36 |
| | **OmniGHA (AR)** | **1.3B** | **35M** | **5.12** |
| | **OmniGHA (AR+Diffusion)** | **1.3B** | **35M** | **4.66** |

Table 2: **Performance on MS-COCO.**

For generation (Table 4), OmniGHA (AR-only) delivers 5.26 images/s, $5.2\times$ faster than Show-o and $4.9\times$ faster than JanusFlow, while remaining close to OmniMamba (5.68 images/s). The AR+Diffusion variant reaches 1.95 images/s—slower due to the diffusion head but still outperforming Show-o and JanusFlow—confirming that the additional cost stems from the generation module rather than the GHA backbone, and that GHA transfers effectively to diffusion pipelines without sacrificing its efficiency advantage in the shared core.

Overall, these results show that GHA not only improves efficiency in AR-only pipelines but also transfers robustly to AR+Diffusion settings, offering a favorable trade-off between speed, accuracy, and extensibility across unified modeling paradigms.

| Ablation | POPE↑ | MME↑ | GQA↑ | FID-30K↓ |
|---|---|---|---|---|
| Linear | 71.8 | 893 | 45.2 | 23.7 |
| Linear + Gate | 80.7 | 1029 | 52.6 | 14.7 |
| Linear + Agent-Softmax | 80.9 | 1048 | 53.2 | 10.5 |
| **GHA** | **82.6** | **1135** | **55.9** | **9.2** |

Table 5: **Attention ablation in AR-only OmniGHA.** We ablate the proposed GHA by starting from a linear attention baseline and incrementally adding its two components: a gating mechanism and an agent-token softmax bottleneck. Both additions individually improve multimodal alignment (POPE, MME, GQA) and image fidelity (FID-30K), while combining them in GHA yields the largest gains. All experiments are conducted in the AR-only setting to isolate the effect of the attention operator.

## 5.4 ABLATION STUDIES

We conduct ablations to isolate the contribution of the proposed GHA attention module, and provide additional results for LoRA configurations, staged pretraining, and agent-token number in Appendix A.6. All ablations are performed in the AR-only setting, as it provides a controlled environment to attribute gains directly to GHA without confounding factors from diffusion modules or complex backbones. For these ablations, we use a lightweight 0.4B model with a reduced agent-token count ($n = 9$) and a lower-cost training setup (shorter schedule and lower image resolution), allowing efficient experimentation while preserving the relative behavior of different attention variants.

Table 5 shows that while the linear attention baseline provides efficiency, it underperforms on multimodal reasoning and generation fidelity. Adding gating alleviates modality imbalance and stabilizes training, while agent-token softmax aggregation restores adaptive global context in long sequences. The full GHA module combines both components, consistently outperforming all partial variants, which confirms that the gains come from the synergy of gating and agent-token aggregation.

To further examine the role of the gating mechanism, we evaluate GHA under intentionally modality-skewed training regimes, where batches contain predominantly text-heavy or image-heavy samples while keeping the agent-token bottleneck fixed. Table 6 summarizes the results. Enabling the gate provides three consistent benefits: (i) it improves performance in both text-skewed and image-skewed settings; (ii) it reduces the performance gap between the two regimes; and (iii) it substantially lowers sensitivity to which modality dominates the data. These findings provide direct evidence that gating mitigates modality imbalance rather than merely improving overall accuracy.

Interestingly, the image-heavy regime even slightly outperforms the baseline one in Table 5 on certain metrics. This phenomenon is also observed in recent studies on modality imbalance, where increasing visual supervision might strengthen cross-modal alignment without degrading overall balance Park et al. (2025). The key point there is that greater exposure to visual signals can reduce the modality gap and enhance multimodal reasoning, rather than harming the model's ability to integrate modalities.

Our focus here is on efficient attention mechanisms under the unified model, but we view the systematic study of modality data distribution shifts as an important and emerging direction for future work.

| Ablation | POPE↑ | MME↑ | GQA↑ | FID-30K↓ |
|---|---|---|---|---|
| GHA w/ gate (more text) | 80.5 | 1059 | 53.5 | 10.8 |
| GHA w/ gate (more image) | 82.9 | 1156 | 56.8 | 10.3 |
| GHA w/o gate (more text) | 78.7 | 1003 | 51.7 | 11.4 |
| GHA w/o gate (more image) | 81.3 | 1094 | 54.4 | 10.9 |

Table 6: **Ablation under modality-imbalanced training.** Gating consistently improves robustness and reduces sensitivity to modality skew.

## 6 CONCLUSION

We introduced GHA, a hybrid operator that augments linear attention with gated recurrence and agent-token–based softmax aggregation, achieving consistent gains over strong unified baselines such as Show-o and JanusFlow while matching the efficiency of OmniMamba. Our study deliberately focused on a controlled setting with moderate scale and simplified AR-only / AR+Diffusion pipelines, ensuring improvements can be attributed directly to the attention mechanism itself. A key limitation is that we have not yet tested GHA within larger-scale unified systems such as BAGEL or Omni-Qwen, which couple massive datasets with more sophisticated paradigms (e.g., mixture-of-experts backbones). We view these as orthogonal and complementary directions, and future work will explore integrating GHA into such advanced frameworks to further assess its scalability and generality. The plug-and-play nature of GHA makes it especially suitable for such integration, offering a lightweight yet expressive operator for the next generation of unified multimodal models.

### REPRODUCIBILITY STATEMENT

We are committed to ensuring the reproducibility of our results. All models were trained using publicly available datasets (MS-COCO, LAION, and GQA) under clearly specified preprocessing steps described in Section 5. We provide detailed hyperparameters, training schedules, and ablation settings in the Appendix. Hardware details, including GPU type, memory footprint, and batch sizes, are explicitly reported in Section 6.2. Random seeds were fixed across all experiments to reduce variance.

For the review phase, we provide an anonymous repository containing the core code for replication purpose: `https://anonymous.4open.science/r/OmniGHA-ICLR2026-63B2`

Upon acceptance, we will release the full codebase, pretrained models, and scripts to reproduce all evaluation results, including ablation studies.

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

# A APPENDIX

## A.1 OMNIGHA MODULES

The modules are shown in Table 7.

Table 7: OmniGHA architecture components and parameter sharing strategy.

| Component | Role | Sharing | Trainable |
|---|---|---|---|
| Text Encoder (SigLIP) | Encode prompts | Shared | Frozen |
| Vision Encoder (DINOv2) | Visual understanding | Task-specific | Frozen |
| Image Tokenizer (LlamaGen) | Visual generation | Task-specific | Frozen |
| GHA Decoder (stacked) | Multimodal fusion | Shared | Trainable |
| Output Heads (linear) | Token/logit mapping | Task-specific | Trainable |
| LoRA Adapters | Task adaptation | Task-specific | Trainable |

## A.2 TRAINING STRATEGY

**Phase 1: Representation Warm-up (Understanding).** We begin by training the shared GHA decoder together with the text head and a task-specific LoRA branch on multimodal understanding tasks. These objectives provide dense token-level supervision, which encourages the gating mechanism to learn stable memory retention and prevents early degeneration.

**Phase 2: Long-context Adaptation (Generation).** Next, we activate the generation-specific LoRA branch and the image head while continuing to update the shared GHA decoder. This phase adapts the gated recurrence to handle long-sequence visual token generation, which exhibits higher variance and requires stable accumulation dynamics.

**Phase 3: Joint Alignment (Unified Fine-tuning).** Finally, we jointly optimize both LoRA branches and modality-specific heads with the shared GHA backbone. Encoders remain frozen. This stage aligns understanding and generation in the shared decoder space, consolidating cross-modal reasoning and ensuring that the gating mechanism generalizes across tasks.

Our design is tailored to the gated–agent structure of GHA. Empirically, we find that direct joint training leads to unstable loss and saturated gates, whereas the proposed progressive schedule provides denser early gradients, stabilizes gating dynamics in long-context generation, and consolidates cross-modal reasoning during fine-tuning. This curriculum not only improves downstream accuracy but also allows us to freeze heavy encoders and concentrate compute on the shared GHA decoder, yielding a more efficient training pipeline.

## A.3 OMNIGHA VARIANTS: AR+DIFFUSION

To demonstrate extensibility beyond the AR-only setting, we also instantiate OmniGHA in an AR+Diffusion variant. In this variant, the shared GHA decoder produces unified multimodal fusion features that are projected via a lightweight latent predictor to form conditioning signals. These conditioning features guide a diffusion-style generation head based on flow matching, which performs efficient velocity field prediction to progressively sample high-fidelity images. The final output image is reconstructed through a pretrained VAE decoder that decodes latent representations into pixels. This design inherits the core principles of the Show-o approach, integrating autoregressive multimodal fusion with a diffusion-based generative mechanism, while demonstrating the flexibility of GHA in supporting diverse unified multimodal pipelines.

## A.4 TRAINING DETAILS AND DATASETS

We constructed our corpus from publicly available multimodal understanding and visual generation datasets and trained the model from scratch without relying on a pretrained backbone. The architecture consists of 48 transformer layers, using a lightweight agent-token configuration of $n = 49$.

For multimodal understanding, DINOv2 and SigLIP encoders are employed (images resized to $384 \times 384$), and a VQVAE tokenizer (LlamaGen) is used for visual generation (images downsampled to $256 \times 256$). LoRA modules (rank 8, adding only 0.65% parameters) were applied to the input projectors of each block. Training was conducted on 64 NVIDIA A100 GPUs in BF16 precision using AdamW ($\beta_1 = 0.9, \beta_2 = 0.95$), cosine annealing with warm-up, weight decay $= 0$, and gradient clipping $= 1.0$.

Our training followed a progressive three-phase curriculum. In *Phase 1 (Representation Warm-up)*, we trained only on multimodal understanding data, including 118K COCO images and 558K samples from the LLaVA-1.5 pretraining set, with batch size 32, learning rate $1 \times 10^{-3}$, 5K steps, and 100 warm-up steps. In *Phase 2 (Long-context Adaptation)*, we switched to text-to-image generation using 35M image–text pairs from CC12M, SA1B, and LAION-aesthetics-12M (excluding MS-COCO 2014), with batch size 90, learning rate $8 \times 10^{-4}$, 100K steps, and 1K warm-up steps. Finally, in *Phase 3 (Joint Alignment)*, we fine-tuned jointly on multimodal understanding (665K LLaVA-1.5 conversations, 220K LVIS-Instruct-4V with GPT-4V instructions, and 400K LRV-Instruct for hallucination mitigation) and text-to-image data, using batch size 48 for generation and 3 for understanding, learning rate $1 \times 10^{-4}$, 150K steps, and no warm-up.

This curriculum provides dense supervision early on, stabilizes gate dynamics during long-context generation, and consolidates cross-modal reasoning during unified fine-tuning while keeping heavy encoders frozen.

## A.5 ADDITIONAL EVALUATION DETAILS

To isolate the architectural contribution of GHA, we train a pure Transformer baseline that mirrors OmniGHA in all aspects of the pipeline. The baseline matches OmniGHA in model size (1.3B), component configuration (Table 7), training corpus, preprocessing, and the full three-phase curriculum. The only difference is that the decoder uses standard multi-head softmax attention instead of GHA.

Table 8 summarizes the understanding and generation results. OmniGHA improves over the Transformer baseline across all metrics (+5.1 POPE, +208 MME-P, +7.3 VQAv2, +1.3 GQA) and reduces FID-30K from 6.15 to 5.12.

| Model | Params | Res. | POPE↑ | MME-P↑ | VQAv2↑ | GQA↑ | FID-30K↓ |
|---|---|---|---|---|---|---|---|
| Standard Transformer | 1.3B | 384 | 83.6 | 1134.7 | 73.3 | 61.1 | 6.15 |
| OmniGHA | 1.3B | 384 | 88.7 | 1342.5 | 80.6 | 62.4 | 5.12 |

Table 8: **Controlled comparison between standard Transformer and OmniGHA.** Both models are trained from scratch under identical data, compute, curriculum, and pipeline setups.

Beyond accuracy, GHA delivers substantial efficiency gains. Table 9 reports inference time across long-context settings. OmniGHA achieves 1.5–7× lower latency for multimodal understanding and 4× higher throughput for image generation.

| Model | 4K | 8K | 16K | 32K | 64K | 128K | Avg. | Img/s | Latency (s) |
|---|---|---|---|---|---|---|---|---|---|
| Standard Transformer | 87 | 104 | 128 | 167 | 204 | 249 | 156 | 1.29 | 15.12 |
| OmniGHA | 12 | 38 | 75 | 110 | 132 | 156 | 87 | 5.26 | 2.97 |

Table 9: **Inference efficiency comparison under identical hardware and decoding settings.**

These results confirm that the performance and efficiency gains of OmniGHA arise directly from the attention operator design rather than differences in model scale, data exposure, or training pipeline. Our Transformer baseline is implemented and tuned under exactly the same pipeline as OmniGHA, and both models are trained from scratch under matched compute and corpus. The empirical improvements therefore reflect architectural behavior rather than confounding factors.

Importantly, our observations are consistent with findings in other hybrid-attention studies. Recent works such as Qwen-Next Qwen.ai (2025) and Kimi Linear Team et al. (2025) (though in different domains) have shown that hybrid attention mechanisms can outperform pure Transformers in certain tasks or regimes, demonstrating that hybrid designs can be competitive and compute-efficient archi-

tectural choices. This contextualizes why, under our unified multimodal setting, GHA can deliver stronger results than the pure-Transformer baseline when budgets are matched.

While full softmax attention is undeniably expressive, unified multimodal decoders face an inherent challenge: modality imbalance. Visual tokens are dense and high-dimensional, whereas text tokens are sparse and low-entropy. Prior multimodal systems such as Flamingo and Show-o introduce additional projections, gating modules, or fusion strategies to stabilize cross-modal interactions under pure Transformers, underscoring this difficulty.

As discussed in Subsection 5.4, GHA targets this issue with two dedicated mechanisms: (1) key–value gating, which balances modality contributions by dynamically normalizing visual vs. textual keys/values; and (2) agent-token softmax aggregation, which stabilizes long-context interactions while preserving the global expressiveness and compensating for strengths associated with pure Transformers. Together, these components form a more stable and compute-efficient architecture for the fixed-budget unified multimodal training regime used in this work, explaining the observed gains over the controlled Transformer baseline and aligning with broader empirical trends in hybrid-attention architectures.

### A.6 ADDITIONAL ABLATION STUDIES

**Impact of Different Unified Model Architecture.**  Table 10 reports the results of different LoRA configurations. Although our backbone is designed to be shared across all tasks, we observe that adding lightweight task-specific components can still bring consistent benefits. This is because the backbone must generalize simultaneously to both dense understanding objectives and sparse generative supervision, which inevitably introduces optimization conflicts. Task-specific LoRA modules act as adapters that specialize the shared decoder to each modality without compromising parameter efficiency.

Removing LoRA modules significantly hurts both multimodal understanding and generation performance (POPE: 80.2, FID: 19.5). Introducing LoRA only for the understanding branch improves understanding metrics (82.1/1069/54.8) but only moderately reduces FID (14.3). Conversely, using LoRA only for the generation branch improves generation (FID: 13.9) while leaving understanding weaker (81.7/1045/54.6). Applying both understanding and generation LoRA leads to consistent gains across all benchmarks (82.6/1135/55.9/9.2), and further adding an output-specific LoRA yields the best performance (82.9/1149/56.3/8.8). These results confirm that task-specific LoRA modules help the model efficiently specialize for each modality while maintaining cross-task synergy.

**Impact of Pretraining Phases.**  Table 11 presents an ablation study of the three-phase pretraining pipeline (Phase 1: representation warm-up, Phase 2: long-context adaptation, Phase 3: unified alignment). Using all three phases yields strong performance (82.6/1135/55.9/9.2), while skipping Phases 1 and 2 and training only with a unified objective leads to clear drops in both understanding and generation (80.9/1046/53.2/12.2).

These observations are consistent with prior unified multimodal systems, where multi-stage training is a standard practice. Pipelines such as Show-o Xie et al. (2024a), JanusFlow Ma et al. (2025), and OmniMamba Zou et al. (2025) all adopt staged curricula, as jointly optimizing understanding and generation with a single decoder is challenging under a single-stage recipe. The curriculum provides benefits to GHA: the warm-up phase supplies dense representation supervision, the long-context phase stabilizes the gate dynamics, and the final alignment phase consolidates cross-modal reasoning.

**Impact of Agent-Token Number.**  We further examine the sensitivity of GHA to the number of agent tokens ($n$). Table 12 summarizes results for $n \in \{4, 9, 16, 25\}$. Performance improves substantially when increasing $n$ from 4 to 9, after which the model enters a stable regime with only marginal gains for $n \in [9, 25]$. This shows that GHA does not rely on a large number of agent tokens—small $n$ already captures most of the benefit—and that the improvements arise from the mechanism itself rather than from a large hidden constant.

To complement accuracy-based analysis, we also measure inference-time scaling at a 64K sequence length. Table 13 shows that runtime grows sublinearly with $n$, confirming that the cost is softened in practice due to our Triton kernel fusion and persistent KV buffer design.

| Ablation | Understanding | | | Generation |
|---|---|---|---|---|
| | POPE↑ | MME↑ | GQA↑ | FID-30K↓ |
| No LoRA | 80.2 | 1031 | 53.3 | 19.5 |
| Only Understanding LoRA | 82.1 | 1069 | 54.8 | 14.3 |
| Only Generation LoRA | 81.7 | 1045 | 54.6 | 13.9 |
| Understanding & Generation LoRA | **82.6** | **1135** | **55.9** | **9.2** |
| Understanding & Generation LoRA + Output LoRA | **82.9** | **1149** | **56.3** | **8.8** |

Table 10: Ablation studies on unified model architecture design.

| Ablation | Understanding | | | Generation |
|---|---|---|---|---|
| | POPE↑ | MME↑ | GQA↑ | FID-30K↓ |
| Stage 1+2+3 | 82.6 | 1135 | 55.9 | 9.2 |
| Only Stage 3 | 80.9 | 1046 | 53.2 | 12.2 |

Table 11: Ablation studies on Pretraining Phases. Phase 1: Understanding Pretraining, Phase 2: Generation Pretraining, Phase 3: Unified Alignment.

| $n$ (agent tokens) | POPE↑ | MME↑ | GQA↑ | FID-30K↓ |
|---|---|---|---|---|
| 25 | 83.6 | 1160 | 58.7 | 8.2 |
| 16 | 82.7 | 1143 | 56.8 | 8.8 |
| 9 | 82.6 | 1135 | 55.9 | 9.2 |
| 4 | 78.4 | 1011 | 50.6 | 12.5 |

Table 12: Ablation on the number of agent tokens $(n)$.

| $n$ (agent tokens) | Inference Time @ 64K (s) |
|---|---|
| 25 | 93 |
| 16 | 78 |
| 9 | 43 |
| 4 | 32 |

Table 13: Inference-time impact of agent-token number.

## A.7   LLM USAGE

This work is entirely original; ChatGPT was used solely for language polishing.