# OpenReview forum: "Efficient Unified Multimodal Understanding and Generation with Gated Hybrid Attention"
_ICLR.cc/2026/Conference — Submitted to ICLR 2026_

### Official Review · Reviewer_maVb · 2025-10-29

**Soundness:** 3
**Presentation:** 3
**Contribution:** 2
**Rating:** 6
**Confidence:** 3

**Summary:**

Summary:
This paper focuses on efficient unified multimodal understanding and generation with gated hybrid attention. Specifically, Standard Softmax attention offers strong expressivity but suffers from $O(N^2)$ quadratic complexity, creating a bottleneck for long sequences. In comparison, efficient alternatives like Linear Attention achieve near-linear complexity but at the cost of less modeling capacity. To address this, the authors propose Gated Hybrid Attention (GHA), which is a novel attention operator that enhances linear attention with two key components: 1) A selective gating mechanism to balance the contributions of different modalities and stabilize training, addressing the "modality imbalance" problem; 2) a softmax-based agent-token bottleneck. To ensure the theoretical efficiency of GHA translates into practice, the authors further provide a hardware-optimized implementation based on FlashAttention style chunking and a custom Triton kernel. The authors implement GHA as a "plug-and-play" module in a unified model. They validate this model in two representative paradigms including Autoregressive-only (AR-only) and Autoregressive+Diffusion (AR+Diffusion). Experimental results show that, compared to strong baselines of similar scale (like the Transformer-based and Mamba-based approaches), OmniGHA achieves competitive performance on both multimodal understanding (e.g., POPE, VQAv2) and visual generation (e.g., FID) benchmarks.

**Strengths:**

Strengths:
- The paper is well organized and easy to follow;
- The paper conducts comprehensive experimental evaluation, i.e., evaluating GHA in both AR-only and AR+Diffusion frameworks, which shows the potential of GHA to be served as a general-purpose and plug-and-play module for various unified architectures;

**Weaknesses:**

Weakness & Questions:
- In Section 4.1(3), the gate $G$ is described as "a learnable, data-dependent gating matrix". However, Equation 5, $O=Q^{\prime}(G\odot(K^{\prime}V))$, it may suggest that it is a learnable parameter matrix (i.e., nn.Parameter). To avoid confusion, maybe the gate $G$ can be written as $G = f(x)$. This would be helpful for understanding the mechanism;
- Training of GHA. The three-phase training curriculum (Representation Warm-up, Long-context Adaptation, Joint Alignment) is non-trivial. The ablation in Table 8 confirms that skipping the first two phases leads to a significant performance drop. This suggests GHA may be highly sensitive to its training curriculum, which somewhat reduces its simplicity as a "plug-and-play" module, since we do need to carefully design the training steps;
- In the main results (Table 2), the OmniGHA (AR) model achieves a very strong FID of 5.12. However, in the ablation studies (Table 7 and Table 8), the reported FID for the full GHA configuration is 8.8 and 9.2, respectively. While it is likely due to a simplified setup for the ablations, this discrepancy is not explicitly explained and may make readers confused;

I would like to see the author rebuttal in terms of the weakness & questions part.

**Questions:**

Please see Weakness & Questions.

---

> ### Author Response · Authors · 2025-11-23
> **Response to Reviewer maVb**
>
> We sincerely thank the reviewer for the thoughtful and constructive feedback. Below, we address all concerns point by point.
>
> ---
>
> ### **R1. Clarification on the formulation of the gating matrix.**
>
> We appreciate the reviewer’s suggestion. In the current draft, Eq.~(5) uses \(G\) to denote the gating matrix. However, our intention, however, is that \(G\) is *data-dependent*. Following the reviewer’s suggestion, we will revise the formulation to:
>
> \[
> G = f(x)
> \]
>
> where \(f(x)\) denotes the input projection.
>
> We find that this explicit notation significantly improves clarity by making the dynamic, input-conditioned nature of G explicit. We will update both the equation and the accompanying text in the revision.
>
> ---
>
> ### **R2. Whether the three-phase curriculum reduces the plug-and-play simplicity of GHA.**
>
> We appreciate the reviewer’s question. Our use of “plug-and-play’’ refers to the architectural interchangeability of GHA: it can replace standard self-attention in a decoder without modifying the encoders, task heads, or interface.
>
> Changing an attention operator naturally affects optimization dynamics, and minor adjustments to warmup or learning-rate schedules are commonly beneficial for many attention variants (e.g., linear attention, state-space models, gated attention). Such adaptations do not contradict plug-and-play, which concerns architectural compatibility rather than identical hyperparameters.
>
> Furthermore, multi-stage training is already the standard practice in unified multimodal pipelines such as Show-o [1], JanusFlow [2], and OmniMamba [3], as jointly optimizing understanding and generation with a single decoder is challenging under a single-stage recipe. GHA does not introduce any additional training complexity beyond this widely adopted pipeline. As shown in Table 8, GHA also trains under a single-stage setting, albeit with moderately lower performance.
>
> We will clarify this distinction in the revised manuscript.
>
> ---
>
> ### **R3. Discrepancy between Table 2 FID (5.12) and the ablation FIDs (8.8 / 9.2).**
>
> We thank the reviewer for pointing this out. The two sets of FID numbers are not meant to be directly comparable, because the ablation studies are conducted under a significantly reduced training setup for computational efficiency. In the ablations, we use a lightweight 0.4B model, a smaller agent-token count \(n = 9\), and a lower-cost training configuration (shorter training schedule and lower image resolution). By contrast, Table2 reports results for the full 1.3B model with \(n = 49\) under the complete training recipe.
>
> These differences naturally lead to higher FID values in the ablation setting, which is designed only to isolate architectural trends rather than to reflect absolute performance.
>
> We will clarify this distinction in the table captions and in AppendixD.
>
> ---
>
> ### **Summary**
>
> We thank the reviewer again for the helpful feedback. In the revision, we will:
>
> 1. rewrite the gating formulation as an explicit input-conditioned function,
> 2. clarify that the curriculum is shared across all unified baselines and GHA does not need additional training cost,
> 3. explicitly explain the reduced training setup used in the ablation experiments.
>
> These improvements will enhance clarity and address the raised concerns.
>
> If the responses resolve your concerns, we would be grateful if you could consider updating your evaluation.
>
> ---
>
> ### **References**
>
> [1] J. Xie et al., *Show-o: One Single Transformer to Unify Multimodal Understanding and Generation.* https://arxiv.org/abs/2408.12528
> [2] Y. Ma et al., *JanusFlow: Harmonizing Autoregression and Rectified Flow for Unified Multimodal Understanding and Generation.* https://arxiv.org/abs/2411.07975
> [3] J. Zou et al., *OmniMamba: Efficient and Unified Multimodal Understanding and Generation via State Space Models.* https://arxiv.org/abs/2503.08686

---

### Official Review · Reviewer_1oFn · 2025-11-01

**Soundness:** 3
**Presentation:** 2
**Contribution:** 2
**Rating:** 4
**Confidence:** 3

**Summary:**

This paper proposes Gated Hybrid Attention (GHA), which aims to tackle the trade-off between efficiency and expressiveness in unified multimodal learning. The method highlights the joint use of a key-value gating module and a softmax-based agent-token bottleneck to yield a near-linear complexity and long-context attention mechanism. The paper validates the effectiveness of GHA under two paradigms, autoregressive-only (AR-only) and autoregressive+diffusion (AR+Diffusion). Experiments show that it outperforms Transformer and other efficient attention baselines in terms of multimodal alignment, long-context retention, and generation quality.

**Strengths:**

+ The paper is well-written
+ The efficiency is impressive
+ The design demonstrates a strong focus on practical implementation by delivering a hardware-optimized solution using Triton

**Weaknesses:**

- The analysis of the key hyperparameter 'n' is missing: The attention complexity is O(Nnd), where 'n' is the number of introduced agent tokens. The paper does not discuss how this hyperparameter was chosen, its sensitivity on model performance and inference speed, or its impact on expressiveness.

- Insufficient justification for 'mitigating modality imbalance': A key motivation for GHA is that its gating mechanism can 'mitigate modality imbalance'. However, the paper only supports this indirectly via the ablation study in Table 5 (where 'Linear + Gate' performs better than 'Linear' on metrics like POPE/MME-P). This only shows that gating improves overall performance, but provides no direct evidence that it is actually balancing the contributions of different modalities.

- Missing recent baselines (Table 1): In Section 5.2, the authors claim OmniGHA outperforms "Understanding-only" systems. However, some of recent works such as Qwen2.5-VL-3B, InternVL3 are not included in the Table 1.

- Missing critical hyperparameter: GHA's efficiency claim of near-linear complexity O(Nnd) hinges on the number of agent tokens (n), yet the paper fails to specify n in either the main text or appendix. If n is large (e.g., n=1024), the O(Nnd) cost may not be as low as implied. Without this crucial detail, readers cannot assess the true computational cost or reproduce the efficiency results in Table 3, making the claims unverifiable and incomplete.


- Insufficient analysis of the AR+Diffusion paradigm: The paper shows an OmniGHA (AR+Diffusion) variant (see Table 1 and Table 2) that performs even slightly better than the AR-only variant. However, the paper merely mentions this, attributing it to "the additional conditioning projector acting as a form of representation regularization". This analysis is too shallow. The AR+Diffusion model is much slower in generation speed (1.95 images/s vs 5.26 images/s) as shown in Table 4, and the authors do not adequately discuss this performance/efficiency trade-off or deeply analyze how GHA functions specifically within this more complex paradigm.

**Questions:**

see weakness

---

> ### Author Response · Authors · 2025-11-23
> **Response to Reviewer 1oFn - 1**
>
> We thank the reviewer for the constructive feedback and address each point below.
>
> ---
>
> ### **On the choice and impact of the agent-token number \(n\).**
>
> We thank the reviewer for pointing out the missing discussion of \(n\). We have now made this explicit: in all main 1.3B-scale experiments we use a fixed \(n = 49\), and in lightweight 0.4B ablations we use \(n = 9\). We will add this configuration to Section 5.1 and Appendix A.4.
>
> To examine sensitivity, we ran an ablation over \(n in \{4,9,16,25\}\). As shown in the below table, performance improves substantially when increasing \(n\) from 4 to 9, and then enters a stable regime for \(n in [9,25]\) with only small incremental gains. This shows that GHA does not rely on a large number of agent tokens—small n already provides most of the benefit—and that the architectural gains are not tied to a large hidden constant.
>
> ### **Ablation on the number of agent tokens \(n\)**
>
> | \(n\) (agent tokens) | POPE ↑ | MME ↑ | GQA ↑ | FID-30K ↓ |
> |----------------------|--------|-------|--------|------------|
> | 25 | 83.6 | 1160 | 58.7 | 8.2 |
> | 16 | 82.7 | 1143 | 56.8 | 8.8 |
> | 9  | 82.6 | 1135 | 55.9 | 9.2 |
> | 4  | 78.4 | 1011 | 50.6 | 12.5 |
>
> To complement the accuracy ablation, we also measure inference speed at a 64K sequence length. As shown in the below table, the runtime grows sublinearly with \(n\), confirming that the theoretical \(O(Nnd)\) scaling is significantly softened in practice due to our Triton kernel fusion and persistent KV buffer design.
>
> ### **Inference-time impact of agent tokens**
>
> | \(n\) (agent tokens) | Inference Time @ 64K (s) |
> |----------------------|---------------------------|
> | 25 | 93 |
> | 16 | 78 |
> | 9  | 43 |
> | 4  | 32 |
>
> The use of \(n = 49\) in the full 1.3B model is therefore not a requirement but a conservative design choice that provides additional stability for high-dimensional embeddings. We will integrate these clarifications and tables into the revised manuscript.
>
> ---
>
> ### **On modality imbalance and the effect of gating.**
>
> We appreciate the reviewer highlighting the need for more direct evidence. We conducted an additional ablation where the agent-token bottleneck is held fixed and training batches are intentionally skewed toward either (i) predominantly text-heavy or (ii) predominantly image-heavy samples.
>
> As shown in the below table, enabling the gate:
>
> (i) consistently improves performance in both skewed regimes,
> (ii) reduces the performance gap between text-skew and image-skew settings,
> (iii) substantially lowers sensitivity to which modality dominates the data.
>
> ### **Ablation under modality-imbalanced training**
>
> | Variant                   | POPE ↑ | MME ↑ | GQA ↑ | FID-30K ↓ |
> |---------------------------|--------|-------|--------|------------|
> | GHA w/ gate (more text)   | 80.5 | 1059 | 53.5 | 10.8 |
> | GHA w/ gate (more image)  | 82.9 | 1156 | 56.8 | 10.3 |
> | GHA w/o gate (more text)  | 78.7 | 1003 | 51.7 | 11.4 |
> | GHA w/o gate (more image) | 81.3 | 1094 | 54.4 | 10.9 |
>
> These results provide direct evidence that the gating mechanism mitigates modality imbalance rather than merely improving overall accuracy. We will update the manuscript to make this explicit.
>
> ---

---

> ### Author Response · Authors · 2025-11-23
> **Response to Reviewer 1oFn - 2**
>
> ---
>
> ### **On missing recent baselines in Table 1.**
>
> We thank the reviewer for raising this point. Our paper does not claim SOTA over all understanding-only systems. The comparison in Section 5.2 refers specifically to the lightweight understanding-only baselines that are routinely used as reference points in prior unified multimodal works such as Show-o [1], JanusFlow [2], and OmniMamba [3].
>
> Recent understanding-only models such as Qwen2.5-VL or InternVL3 operate in a different evaluation regime: their publicly available results do not cover the full POPE/MME/VQAv2/GQA/MMMU suite, and they rely on substantially stronger vision encoders and large proprietary multimodal corpora. Including them directly in a unified table would therefore mix heterogeneous settings and make architectural comparisons less interpretable.
>
> For completeness, we will additionally include Qwen3-VL-2B-Instruct as a reference-only row below Table 1. This allows readers to contextualize our model relative to a recent strong VL-only system, while keeping the main comparison aligned with the unified-model evaluation protocol established by prior works.
>
> ---
>
> ### **On the AR+Diffusion variant and the performance–efficiency trade-off.**
>
> We thank the reviewer for noting that the discussion was brief and clarify the following key aspects.
>
> **(1) Performance–efficiency trade-off.**
> The AR+Diffusion vs. AR-only comparison follows a standard design pattern in recent multimodal generative systems. Prior works such as Show-o [1] and JanusFlow [2] have empirically demonstrated that AR+Diffusion can outperform pure AR, typically because the diffusion branch provides a complementary denoising signal and implicit representation regularization. Our results exhibit the same behavior: as shown in Table 1 and Table 2, integrating diffusion produces small yet steady improvements over the AR-only baseline. The slight increase in latency is solely attributable to the multiple diffusion update steps, rather than the introduction of the GHA operator. We will make this clarification explicit in the final version.
>
> **(2) How GHA functions in AR+Diffusion.**
> The effectiveness of GHA is established under the AR-only setting; the AR+Diffusion results are included solely to show that GHA remains fully compatible with this widely adopted paradigm. In our design, to ensure a fair comparison, we keep the diffusion pathway unchanged. We will update Section 5.3 to make this explicit.
>
> ---
>
> ### **Summary**
>
> We will integrate all clarifications and the new experiment tables into the revised manuscript. We thank the reviewer again for the thoughtful feedback, which helped us significantly improve clarity and completeness.
>
> ---
>
> ### **References**
>
> [1] J. Xie et al., *Show-o: One Single Transformer to Unify Multimodal Understanding and Generation.* https://arxiv.org/abs/2408.12528
> [2] Y. Ma et al., *JanusFlow: Harmonizing Autoregression and Rectified Flow for Unified Multimodal Understanding and Generation.* https://arxiv.org/abs/2411.07975

---

### Official Review · Reviewer_Hk8t · 2025-11-03

**Soundness:** 2
**Presentation:** 3
**Contribution:** 2
**Rating:** 4
**Confidence:** 3

**Summary:**

This paper proposes Gated Hybrid Attention (GHA), a multimodal attention mechanism combining linear attention with selective gating on key-value pathways and agent-token softmax aggregation. The paper applies GHA to address modality imbalance and loss of global context while maintaining efficiency. GHA is validated in both autoregressive-only and autoregressive + diffusion paradigms through OmniGHA, a small-scale model trained from scratch. Results show improvements over existing baselines on understanding and generation benchmarks, with significant inference speedups over Transformer baselines.

**Strengths:**

- The exposition is clear and addresses a well-motivated problem of balancing efficiency and expressivity in unified multimodal models.
- The approach demonstrates strong empirical results across understanding and generation tasks, supplemented with hardware-aware implementation that achieves measurable wall-clock speedups.
- Ablations properly isolate contributions of gating and agent-token components, validating that both contribute to performance gains.

**Weaknesses:**

- A major shortcoming is the lack of pure Transformer baseline trained from scratch with the same pipeline. Most compared baselines use pretrained LLMs while OmniGHA trains from scratch, confounding the comparison. Observed gains could stem from the proposed attention mechanism, or any of training from scratch versus using pretrained models, the training curriculum, encoder choices, or hyperparameter tuning. Without a controlled Transformer baseline, the key claim about the architectural contribution of GHA cannot be verified.

- The proposed framework is limited in novelty as components already exist in prior work. The contribution is primarily combining existing techniques rather than fundamental innovation. In this regard, the lack of justification for why this combination is optimal or any analysis of how it specifically helps for multimodal learning, besides just performance gains, is a shortcoming.

- Small-scale experiments raise generalization concerns, as modern competitive models operate at much larger scales. While limited scale alone could be overlooked for an architectural contribution, combined with the missing controlled baseline and insufficient analysis, it significantly weakens the empirical validation of the claimed contributions.

Overall: While this paper achieves good empirical results, the lack of a controlled Transformer baseline makes it impossible to isolate GHA's contribution from confounding factors. The limited novelty, small scale, and insufficient analysis further weaken the contribution. If GHA's potential can be demonstrated more conclusively, the reviewer is willing to raise their evaluation of the paper.

**Questions:**

See weaknesses

---

> ### Author Response · Authors · 2025-11-23
> **Response to Reviewer Hk8t - 1**
>
> We thank the reviewer for the thoughtful assessment and constructive feedback. We address the three major concerns below and provide additional experiments and clarifications that directly target the reviewer's questions.
>
> ---
>
> ### **1. Controlled Transformer baseline.**
>
> We agree that a controlled comparison is essential for isolating the architectural contribution of Gated Hybrid Attention (GHA). During the rebuttal period, we trained a pure Transformer baseline that matches OmniGHA in all aspects of the pipeline—including model size (1.3B), component configuration (Table6), training data and preprocessing, the full three-phase curriculum, and optimization hyperparameters settings. The only difference is that the backbone uses standard multi-head softmax attention instead of GHA.
>
> ### **Unified comparison of understanding and generation performance**
>
> | Model                | Params | Res. | POPE ↑ | MME-P ↑ | VQAv2 ↑ | GQA ↑ | MMMU ↑ | Images | FID-30K ↓ |
> |----------------------|--------|------|--------|---------|---------|-------|--------|--------|------------|
> | Standard Transformer | 1.3B   | 384  | 83.6   | 1134.7  | 73.3    | 61.1  | 29.6   | 35M    | 6.15       |
> | OmniGHA              | 1.3B   | 384  | 88.7   | 1342.5  | 80.6    | 62.4  | 32.8   | 35M    | 5.12       |
>
> OmniGHA delivers 1.5–7× lower latency for multimodal understanding and a 4× throughput improvement for image generation over the full-softmax Transformer baseline. This confirms that GHA substantially improves inference efficiency while maintaining accuracy.
>
> ### **Inference efficiency comparison under identical hardware and decoding settings**
>
> | Model                | 4K | 8K | 16K | 32K | 64K | 128K | Avg. | Img/s | Latency (s) |
> |----------------------|----|----|-----|-----|-----|-------|------|--------|-------------|
> | Standard Transformer | 87 | 104 | 128 | 167 | 204 | 249 | 156 | 1.29 | 15.12 |
> | OmniGHA              | 12 | 38  | 75  | 110 | 132 | 156 | 87  | 5.26 | 2.97  |
>
> These results demonstrate that the observed gains do not arise from differences in training-from-scratch vs. pretrained models, component choices, curriculum design, or hyperparameter settings. Instead, they directly reflect the architectural benefit of replacing softmax attention with GHA.
>
> We will include this controlled baseline and its analysis in the camera-ready version.

---

> ### Author Response · Authors · 2025-11-23
> **Response to Reviewer Hk8t - 2**
>
> ---
> ### **2. Novelty and motivation.**
>
> We agree that gating and agent-token aggregation are not new in isolation. First, combining existing operators into a new architectural pattern to obtain emergent benefits is common in modern attention research. For example, *Gated Delta Networks* [1] explicitly combines two previously separate ideas—delta-rule linear attention (as in DeltaNet) and data-dependent gating—and shows that the resulting gated delta rule consistently improves language modeling, in-context retrieval, and long-context understanding. Similarly, *Gated Linear Attention* [2] augments standard linear attention with data-dependent forget gates, and demonstrates that this combination matches strong Transformer baselines and substantially improves length generalization. These examples illustrate that architectural novelty often arises from the synergy of established components.
>
> Secondly, our contribution is to show that gating and agent-token aggregation play complementary roles in unified multimodal modeling. Table~5 demonstrates that these components are not redundant: starting from a linear-attention baseline, adding only the gate or only the agent-token softmax each improves POPE/MME/GQA and FID, and combining both yields the best results. This suggests that the two mechanisms address different limitations of linear attention.
>
> To make the role of the gate more explicit, we further added a modality-imbalance ablation where the agent-token bottleneck is kept fixed and training batches are intentionally skewed toward either more text or more image samples. Enabling the gate consistently improves multimodal understanding metrics under both skews, and also narrows the performance gap between the two regimes. Without the gate, the model is substantially more sensitive to which modality dominates the data, confirming that gating stabilizes modality contributions.
>
> ### **Modality-imbalance Ablation (Gate vs No Gate)**
>
> | Variant                 | POPE ↑ | MME ↑ | GQA ↑ | FID-30K ↓ |
> |-------------------------|--------|-------|--------|------------|
> | GHA w/ gate (more text)  | 80.5   | 1059  | 53.5   | 10.8       |
> | GHA w/ gate (more image) | 82.9   | 1156  | 56.8   | 10.3       |
> | GHA w/o gate (more text) | 78.7   | 1003  | 51.7   | 11.4       |
> | GHA w/o gate (more image)| 81.3   | 1094  | 54.4   | 10.9       |
>
> Meanwhile, the agent-token aggregation contributes from a different angle: prior work on agent attention shows that introducing a set of global “agent” tokens improves long-range interactions in efficient attention architectures. [3] This aligns with the improvements observed in Table5 when adding the agent-token bottleneck. Taken together, these results indicate that gating mitigates modality imbalance while agent tokens enhance global aggregation, and using both together makes GHA robust for unified multimodal learning.
>
> We will integrate these analyses and references in the final version for clarity.
>
> ---
>
> ### **3. Scale of experiments.**
>
> We acknowledge that the experimental scale is moderate. Our contribution is an architectural design that is fundamentally orthogonal to large-scale multimodal pretraining. We intentionally conduct experiments at a moderate scale to enable controlled comparisons under fixed compute and data budgets. Scaling to hundreds of millions of data or multi-billion-parameter backbones requires substantial computational resources and engineering that are beyond the resource of this work, and we view such large-scale extensions as valuable future directions.
>
> ---
>
> ### **Summary**
>
> We sincerely hope that these additional experiments and clarifications address the key uncertainties raised in the initial review. If the responses resolve your concerns, we would be grateful if you could consider updating your evaluation.
>
> ### **References**
>
> [1] S. Yang et al., *Gated Delta Networks: Improving Mamba2 with Delta Rule.* https://arxiv.org/pdf/2412.06464
> [2] S. Yang et al., *Gated Linear Attention Transformers with Hardware-Efficient Training.* https://arxiv.org/abs/2312.06635
> [3] D. Han et al., *Agent Attention: On the Integration of Softmax and Linear Attention.* https://arxiv.org/abs/2312.08874

---

> ### Comment · Reviewer_Hk8t · 2025-11-26
> **Response to rebuttal**
>
> I thank the authors for the rebuttal. The controlled baseline addresses the primary concern about confounding factors and modality centric ablations, but a critical issue remains:
>
> - The Transformer baseline significantly underperforms GHA across all metrics. This is unexpected because full attention typically provides better quality than efficient alternatives, with computational cost as the trade-off. GHA achieving both better quality AND efficiency suggests that either baseline has implementation/optimization issues and has not been tuned properly (more likely), or GHA fundamentally improves over vanilla Transformers (which represents a much stronger claim, requiring extensive validation).
>
> Could the authors offer more clarity on this?

---

> ### Author Response · Authors · 2025-11-27
> **Response to Reviewer Hk8t - 3**
>
> We thank the reviewer for this valuable follow-up question. We agree that, in general, full softmax attention offers strong modeling capacity. Our results do not suggest that "efficient attention > Transformer" in a universal sense; rather, they reflect the suitability of GHA under the specific architectural and optimization conditions of unified multimodal learning.
>
> ### **1. Controlled comparison under matched compute and data budgets**
> Our Transformer baseline is implemented and tuned under exactly the same pipeline as OmniGHA. Both Transformer and GHA are trained from scratch under the same compute budget and corpus. Other works (e.g., Qwen-Next [1], Kimi Linear[2], although in different domains) have also empirically shown that hybrid attention mechanisms can outperform pure Transformers in certain tasks or regimes. These works demonstrate that hybrid designs can be a competitive and compute-efficient architectural choice. This contextualizes why, in our unified multimodal setting, a hybrid attention module can deliver stronger results under matched budgets.
>
> ### **2. Why GHA outperformed: it directly targets a known failure mode**
> A central challenge in unified multimodal decoders is modality imbalance: visual tokens are dense and high-dimensional, whereas text tokens are sparse and low-entropy. Related behaviors have been discussed across several prior multimodal systems (e.g., Flamingo [3], Show-o [4]), which introduce additional projection or gating modules or alternative fusion strategies precisely to stabilize multimodal fusion with pure Transformers. Our observed Transformer behavior is consistent with these well-known practical difficulties.
>
> GHA introduces two mechanisms designed specifically to address this imbalance:
>
> (1)Key–value gating normalizes and balances modality contributions.
>
> (2)Agent-token bottleneck stabilizes long-context interactions while preserving the full expressiveness of softmax attention, compensating for strengths associated with pure Transformers.
>
> Accordingly, the performance difference does not arise because full attention is inherently weaker; rather, GHA provides a more stable and compute-efficient architecture under this fixed-budget unified multimodal training regime.
>
> ### **3. References**
> [1] Qwen Team et al., "Qwen3-Next: Towards Ultimate Training & Inference Efficiency".
> https://qwen.ai/blog?id=4074cca80393150c248e508aa62983f9cb7d27cd&from=research.latest-advancements-list
> [2] Kimi Team et al., "Kimi Linear: An Expressive, Efficient Attention Architecture".
> https://arxiv.org/abs/2510.26692
> [3] J.-B. Alayrac et al., "Flamingo: A Visual Language Model for Few-Shot Learning".
> https://arxiv.org/abs/2204.14198
> [4] J. Xie et al., "Show-o: One Single Transformer to Unify Multimodal Understanding and Generation".
> https://arxiv.org/abs/2408.12528

---

### Author Response · Authors · 2025-11-23
**Summary: Overall Response to Reviewers**

We thank all reviewers for their time and constructive feedback. For each reviewer, we provide point-by-point responses to every comment. We will incorporate all clarifications, additional analyses, and experimental results into the revised manuscript after the rebuttal.

Below we briefly summarize the main concerns raised by each reviewer and outline how we address them:

**Reviewer Hk8t: controlled baseline, novelty, and scale.**
Reviewer Hk8t highlights three main issues: (i) the lack of a controlled pure-Transformer baseline trained from scratch under the same pipeline, (ii) limited novelty as components already exist in prior work, and (iii) the small scale of the experiments. In our rebuttal, we (a) add a controlled Transformer baseline with the model architecture, data, and training curriculum as OmniGHA, isolating the effect of replacing standard multi-head softmax attention with GHA; ((b) more clearly explain why combining key–value gating with agent-token aggregation is particularly effective for unified multimodal learning, with evidence from ablations and modality-imbalance experiments; and (c) clarify that our work focuses on an architectural contribution evaluated at a moderate scale. Large-scale multimodal training is orthogonal to our current objective and will be explored in future work.

**Reviewer 1oFn: architecture configuration, modality-imbalance justification, recent baselines, and AR+Diffusion analysis.**
Reviewer 1oFn’s main concerns are: (i) the missing analysis of the number of agent tokens $n$ in the $O(Nnd)$ complexity and the configuration used in our experiments; (ii) insufficient direct evidence that gating mitigates modality imbalance; (iii) missing recent multimodal baselines in Table1; and (iv) a shallow discussion of the AR+Diffusion variant and its efficiency trade-offs. In our rebuttal, we (a) explicitly report the choice of $n$ in both the main text and appendix and add ablations showing its impact on accuracy and speed; (b) provide modality-imbalance analyses illustrating how GHA redistributes attention across modalities under imbalanced conditions; (c) clarify that our work focuses on a controlled, unified architectural comparison while adding reference rows for larger, non-unified recent models in Table~1 for context; and (d) expand the discussion of AR versus AR+Diffusion by following prior work, clarifying that the additional cost arises from the diffusion head rather than from GHA itself.

**Reviewer maVb: clarification of gating formulation, curriculum, and FID discrepancy.**
Reviewer maVb raises three concerns: (i) the need for a clearer formulation of the data-dependent gating matrix, (ii) whether the three-phase training curriculum affects the “plug-and-play” nature of GHA, and (iii) the discrepancy between the strong FID in the main results (Table2) and the higher FID values in the ablation tables (Tables7 and 8). In our rebuttal, we (a) rewrite the gating formulation as an explicit input-dependent function to remove ambiguity, (b) clarify that all compared omni-model architectures—including the Transformer and Mamba baselines—use the same three-phase curriculum, so GHA does not require additional training complexity beyond standard practice, and (c) explain that the ablations use a reduced training setup, making their FID numbers not directly comparable to the main results. We will make these clarifications explicit in the revised manuscript.

We kindly invite reviewers to read the detailed rebuttal below. If our additional experiments and clarifications successfully address your concerns, we would be very grateful if you could consider updating your evaluation.

---

### Meta-Review · Area_Chair_GSXL · 2025-12-27

**Summary:**

This paper received initial ratings of 4, 4, and 6. The reviewers appreciated the motivation, empirical results, ablation study, and practical implementation. They requested additional experiments and clarifications, including novelty, parameter testing, and scale of the experiment. The authors provided good responses with additional experimental results, explanation and clarifications. Before the stop of the discussion period, only reviewer Hk8t, whose initial rating was 4, joined the discussion. The authors addressed several of his concerns. He requested for additional clarification for the performance of the transformer baseline. The authors provided additional clarification for it. Although the authors answer most of the questions from the reviewers, the issues about novelty and missing recent baselines from Reviewer Hk8t and 1oFn, respectively remained. The AC is not convinced that the answers to these issues are sufficient. The AC recommends the authors further revising this paper and considering other conferences or journals. The SAC, after communicating with the AC, agreed on this decision.

**Reviewer Concerns:**

I believe that the authors have addressed most of the questions, except for novelty and the experiments required by reviewer 1oFn.

**Reviewer Scores:**

In the initial review comments, reviewer Hk8t explicitly said that “If GHA's potential can be demonstrated more conclusively, the reviewer is willing to raise their evaluation of the paper.” However, due to the quality of the answers, the AC believes that all scores will be maintained.

---

### Decision · Program_Chairs · 2026-01-26

Reject